# A Contribution to Improve Barrier Properties and Reduce Swelling Ratio of κ-Carrageenan Film from the Incorporation of Guar Gum or Locust Bean Gum

**DOI:** 10.3390/polym15071751

**Published:** 2023-03-31

**Authors:** Ruixuan Wang, Song Zhang, Shuaichen Liu, Yuqi Sun, Hongyan Xu

**Affiliations:** Department of Food Science and Engineering, College of Agriculture, Yanbian University, Yanji 133000, China; 13945977105@163.com (R.W.);

**Keywords:** κ-carrageenan, galactomannan, composite film, physical properties

## Abstract

In this study, galactomannan (GM), including guar gum (GG) or locust bean gum (LG), was incorporated into a κ-carrageenan film to improve barrier properties and reduce the swelling ratio (SR). The effects of that with different concentrations on optical, mechanical, barrier, swelling and thermal properties of the κ-carrageenan-based film were researched. SEM and rheological results showed that both κ-carrageenan/GG and κ-carrageenan/LG had good compatibility and stability. FTIR results showed that LG was easier to form hydrogen bonds with κ-carrageenan. The KC-L exhibited excellent mechanical properties, barrier properties, and SR than KC-G. The film with 15% GM had good light transmittance. Moreover, the thermal stability of the film could be improved by adding GMs. This study reports that the κ-carrageenan–GM film has potential in packaging applications.

## 1. Introduction

Plastic has caused a series of serious environmental issues. Every year, about 500 billion plastics bags were used, but only 3% of them could be recycled. The rest of those were discarded as solid waste, which is difficult to degrade since it causes white pollution to the ecological environment and landscape. In addition, plastics have a potential threat to human health because they may contain some chemical additives such as bisphenol A or phthalates. Thus, it is urgent to find a sustainable alternative [1]. Natural polysaccharides have received increased attention due to their renewability, biodegradability and film-forming properties [2]. Marques et al. prepared cassava starch-based films that can be used for packaging in the food industry [3]. The calcium alginate film loading ciprofloxacin potentially healed infected foot ulcers [4]. Chargot et al. fabricated a chitosan/carrot cellulose nanofibre film with antibacterial properties [5]. It is noted that polysaccharide films could effectively block the transport of CO_2_ and O_2_, decreasing the quality loss of foods during storage. Carrageenan, a kind of linear polysaccharide with negative charges, is widely used in the field of food [6]. It has been used as the main film-forming substance, and has been turned into antimicrobial packaging film [7] and freshness monitoring labels [8]. However, carrageenan-based films swell easily in moist conditions [9] and have poor water vapor barrier properties [10,11]. Shahbazi et al. blended high-pressure homogenized starch into a carrageenan film to overcome those drawbacks. However, it is troublesome for starch to be modified physically [12]. Hanani et al. added various plant oils into κ-carrageenan films to reduce WVP, while their mechanical properties decreased significantly [13]. Nanosilica was added to the κ-carrageenan film to improve the performances. However, dynamic oscillation measurements showed that the dissipation factor decreased significantly with the angular frequency from 0.1 to 100 rad/s, indicating that the stability of the film solution was poor [14].

Galactomannans consisting of a mannan (M) backbone with single galactose (G) are commonly found in the endosperm of legumes [15] and have been widely used for a vast range of foods [16], packaging [17] and pharmaceuticals [18]. Three major commercial galactomannans include guar gum, tara gum and locust bean gum, and the corresponding ratios of M/G were 2:1, 3:1 and 4:1, respectively. Figure 1 shows the typical structure of galactomannan (idealized) [19]. Wu et al. discovered that tara gum (10% to 60%) resulted in a higher gel strength of mixed gels than that of pure κ-carrageenan [20]. JT et al. prepared a clay/κ-carrageenan/LG composite film with potential in food packaging and presenting good mechanical properties, lower WVP and better antibacterial activity for L. monocytogenes at the addition of 16 wt% clay [21]. As far as we know, the effects of galactomannan with different ratios of M/G and content on inherent properties of κ-carrageenan films have not been reported.

In this study, we prepared a composite films by mixing κ-carrageenan and GM solutions in which glycerin was used as a plasticizer to solve the drawback of brittleness. The effects of GM with two ratios of M/G (GG and LG) and the content on κ-carrageenan-based films were investigated. We used scanning electron microscopy (SEM) to observe the cross sections of the films and employed Fourier-transform infrared (FTIR) and thermogravimetric analysis (TGA) to research the functional groups and thermal stability. The light transmittance, mechanical properties, oxygen permeability, water vapor permeability, swelling, and rheological properties of ĸ-carrageenan–GM composite films were also investigated.

## 2. Materials and Methods

### 2.1. Materials

Guar gum was bought from Fiyyed Biotech Co., Ltd. (Suzhou, China). Locust bean gum was purchased from Yaheng Biotech Co., Ltd. (Heze, China). κ-Carrageenan (food grade) was supplied by Bairen Biotech Co., Ltd. (Qingdao, China). Glycerol (AR) was offered by Yongda Chemical Reagent Co., Ltd. (Tianjin, China). Ethanol (AR) was obtained from Fuyu Chemical Co., Ltd. (Tianjin, China).

### 2.2. Preparation of Composite Film

GG or LG (5, 10, 15, 20, and 25%, *w*/*w*, based on κ-carrageenan and GM) was firstly dispersed in 10 mL of ethanol and then stirred in distilled water at 65 °C. At the same time, κ-carrageenan was dissolved in distilled water under stirring at 85 °C. After that, the solution was added into GM solution and stirred for 60 min. Subsequently, 30% (*w*/*w*, based on κ-carrageenan and GM) of glycerin was added and stirred for another 30 min. Finally, the resulting film-forming solution was cast into a mold (300 mm × 290 mm) and dried for ~24 h at 50 °C. The corresponding films were labeled based on the concentrations of GG and LG. For example, the composite film consisting of 5% GG was recorded as KC-5G. The films prepared from 30% glycerol and GG or LG are noted as GGF and LGF, respectively.

### 2.3. Characterizations

The cross-sections of the composite films were observed using a Quanta 200 SEM (Philips-FEI Co., Eindhoven, The Netherlands). The functional groups were studied using a Nicolette 6700 spectrometer (Perkin Elmer, Frontier, MA, USA) with attenuated total reflection mode in the range of 600–4000 cm^−1^ at a resolution of 4 cm^−1^.

### 2.4. Film Properties

#### 2.4.1. Light Transmittance

The light transmittance of the films (2.5 cm × 4.0 cm) was measured using an UV-2600 spectrophotometer (Shimadzu, Kyoto, Japan) in the range of 200–800 nm.

#### 2.4.2. Mechanical Properties

Thickness of the film was obtained by measuring 20 random points using a micrometer (ID-C112XBS, Mitutoyo Corp., Tokyo, Japan). An XLW-PC auto tensile tester (Labthink, Jinan, China) was used to measure tensile strength (TS) and the elongation at break (EB) of the films (1.5 cm × 8.0 cm) with a strain rate of 300 mm/min.

#### 2.4.3. Oxygen Permeability

The oxygen permeability (OP) of the film was measured using an OX/230 oxygen transmission rate tester (Labthink Instruments Co., Ltd., Jinan, China).

#### 2.4.4. Swelling Property

The swelling property of the films was determined according to a modified method [22]. The films (15 mm × 15 mm) were dried and the volume of them was measured, then they were placed in separate dryers with relative humidities of 53% and 75% for 24 h at room temperature. The swelling ratio of the films was calculated using the following formula:(1)$$\mathrm{SR}=\left(v_{s}-v_{e}\right)/v_{e}$$
where $v_{s}$ and $v_{e}$ represent the volume of the films before and after swelling, respectively.

#### 2.4.5. Water Vapor Permeability

The water vapor permeability (WVP) of the films was obtained through the gravimetric method. The weighing bottles containing 23 g of anhydrous calcium chloride were sealed with the film (~16.6 cm^2^) then conditioned in desiccators under 75% RH at 25 °C. The weight was weighed periodically, and the WVP was calculated according to the following equation:$$\mathrm{WVP}=\left(k\times d\right)/\left(s\times \Delta P\right)$$
where k is the weight of moisture gain per unit of time (g/s), d is the average film thickness (mm), s is the area of the exposed film surface (m^2^) and ΔP is the driving force (1753.55 Pa).

#### 2.4.6. Thermogravimetric Analysis

TGA Q500 (TA Instruments, DE, USA) was used to characterize the thermal stability in a nitrogen atmosphere from room temperature to 600 °C with a heating rate of 10 °C/min. The derivative thermogravimetric analysis (DTG) was obtained via the first-order derivative of TGA.

#### 2.4.7. Rheological Analysis

After the films were immersed in distilled water for 30 s, the rheological behavior of the films with different concentrations of GG and LG in gel state was measured by using a rotational rheometer (TAAR2000EX, Newcastle, DE, USA) with a parallel-plate geometry (diameter = 40 mm, gap = 1000 μm) at a shear rate range of 0.1–100 s^−1^ at 25 °C. The rheological properties of films in the gel state were studied by using the Ostwald–de Wale model, as below:(2)$$\eta_{a}=Κ\times \gamma^{n-1}$$
where $\eta_{a}$ is the viscosity (Pa•s), γ denotes the shear rate (s**^−^**^1^), n represents the flow index and K is the parameter of the materials.

The viscoelasticity of the films in gel states was characterized by dynamic oscillation measurements in the frequency range of 0.1–100 rad/s. The viscoelasticity of the films in gel states can be described via the following equation:(3)$$tan\delta =G^{\prime \prime }\left(\omega \right)/G^{\prime }\left(\omega \right)$$
where tanδ, $G^{\prime \prime }\left(\omega \right)$ and $G^{\prime }\left(\omega \right)$ are the dissipation factor, loss modulus and the storage modulus, respectively.

#### 2.4.8. Statistics Analysis

Data were analyzed by SPSS Statistics (v17.0; SPSS Inc., Chicago, IL, USA). The differences among mean values were processed by using the Duncan multiple range tests (*p* < 0.05).

## 3. Results and Discussions

### 3.1. SEM Analysis

Micrographs of the cross-sections of κ-carrageenan-based films containing various concentrations of GG and LG are exhibited in Figure 2. An ordered and homogeneous stripy structure exhibited on the cross-section of the κ-carrageenan-based film can be seen due to the helical aggregation of κ-carrageenan chains; meanwhile, the entanglements among the κ-carrageenan and the GG/LG molecule chains hint at a good compatibility and interactions between GM and κ-carrageenan. In addition, it can be observed that the cross-section of the LG–carrageenan film was more compact than that of the GG–carrageenan film. This is because the branch units of the GG are twice that of the LG, thus the compatibility of the GG–carrageenan film is lower.

### 3.2. FTIR Spectroscopy

The FTIR spectra of KC, KC-25G and KC-25L are shown in Figure 3. The bands at 3301 cm^−1^, 2925 cm^−1^, 2884 cm^−1^ and 1022 cm^−1^ are assigned to O-H stretching vibration, C-H stretching vibration, C-H bending vibration, and C-O (pyranose ring) stretching vibration, respectively [23,24]. For KC, the bands at 1239 cm^−1^ (sulfate ester), 918 cm^−1^ (3,6-anhydrogalactose) and 846 cm^−1^ (galactose-4-sulfate ester) are the characteristic peaks of κ-carrageenan [25]. After the incorporation of GG and LG, the spectra of KC-25L and KC-25G shows the bands at 868 cm^−1^ and 814 cm^−1^, which are attributed to the β-linked D-mannopyranose units and α-linked D-galactopyranose units, respectively [26]. Meanwhile, the two spectra of KC-25L and KC-25G are similar. However, it is found that the intensity of the band at 3295 cm^−1^ decreased and shifted to 3280 cm^−1^, which is owing to the fact that with LG with fewer branched chains it is easier to form hydrogen bonding because of the lower steric hindrance. Martins et al. also found that carrageenan could interact with the unbranched smooth segments of LG via hydrogen bonds [26].

### 3.3. Optical Properties

The optical properties of the composite films are shown in Figure 4. With the increase in GM, the light transmittance of the composite films increased first and reached the maximum value with a 15% GM addition, then decreased. The regularity of KC-L was better than that of KC g due to the fewer branch chains of LG because the scattering and refraction of light decreased. When the GM content was less than 15%, the transmittance of KC-L was poor compared with that of KC-G. When the GM content further increased from 15% to 25%, the transmittance of KC-L and KC g decreased from 57.06% to 51.74% and 54.58% to 43.24%, respectively, showing that the light transmittance of KC-L was slightly higher than that of KC-G. This is mainly because the looser and more disordered network of the composite film formed after adding the GG with more branched chains, resulting in more reflection and refraction when light passed through the films.

### 3.4. Mechanical Properties

The mechanical properties of the κ-carrageenan film and GM–κ-carrageenan film are displayed in Table 1. EB decreased when the GM content increased from 5% to 25% because the substituted and branched structure of the GM obstructed the formation of intermolecular hydrogen bonds, resulting in an easier breaking (lower EB). When the content of GG was in the range of 5–15%, the TS of KC g had no obvious change. This is because GG with a higher branching degree is difficult to form more hydrogen bonds with κ-carrageenan. It can be seen in Figure 5a that GG just interweaved randomly with κ-carrageenan. The TS of KC-5L and KC-10L was higher than that of KC due to the formation of hydrogen bonds between the unsubstituted regions of the LG chain and κ-carrageenan (Figure 5b) [27]. In addition, the larger steric hindrance of GM tended to hinder the formation of intermolecular hydrogen bonds, so the TS value gradually decreased with the further increase in the content.

### 3.5. Oxygen Permeability

The OP of GM–κ-carrageenan films are shown in Table 1. The addition of GM could effectively reduce the gap through random interleaving and winding, resulting in a lower OP of KC-5G and KC-5L than that of the KC film (3.247 ± 0.490cm^3^ mm m^−2^ day^−1^ atm^−1^). The KC g with 20% GG and KC-L with 15% LG had the minimum value of OP. When the amount of GM continued to increase, the density of the composite film decreased again. A similar result was obtained with the chitosan and guar gum composite films [28]. Furthermore, the OP of KC-L was lower than that of KC g under the same addition because the fewer branch chains of LG made the film structure denser.

### 3.6. Swelling Property and WVP

The SR and WVP of the κ-carrageenan-based composite films are displayed in Table 2. The SR of the κ-carrageenan-based composite films decreased with the increased GM content and decreased relative humidity. This is because the wettability of κ-carrageenan decreased as the GM content increased, leading to the decrease in water transfer (WVP). A similar result was achieved when the WVP of the yellow tragacanth-LG composite film decreased with the increase in LG [29]. In addition, the SR and WVP of KC g were higher than that of KC-L with the same content because GG with a higher branching degree decreased the compactness of the composite films. Kurt and Kahyaoglu also reported that the WVP of GG films was higher than that of LG films [30].

### 3.7. Thermal Stability

TGA and DTG curves of KC-GM films are shown in Figure 6. The decomposition parameters for GG, LG and all samples are listed in Table 3. For the KC films, the first stage was from 21.70 to 121.47 °C due to the evaporation of moisture. The second one from 121.47 to 227.43 °C was related to the pyrolysis of glycerol, and the third weight loss at 227.43~414.91 °C was attributed to the decomposition of polysaccharide. A similar thermal decomposition curve was observed with the cassia gum film added with a plasticizer [31]. The thermal decomposition temperature of the GM–κ-carrageenan composite films was higher than that of the control films because the thermal stability of neutral polysaccharide is higher than that of the polysaccharide with charges. The pyrolysis temperature of GGF was higher than that of LGF, which resulted in a lower thermal stability for KC-L.

### 3.8. Rheological Analysis

Effects of galactomannan concentrations on steady-state shear properties of the films in gel states are shown in Figure 7a,b. The viscosity of the gel decreased with the shear rate range from 0.1 to 100 s^−1^, which showed a pseudoplastic fluid. This shear-thinning behavior might be caused by the inability of the intermolecular entanglement structure to recombine under shear. The Ostwald–de Wale model was used to research the relationship between the shear rate and the apparent viscosity. As shown in Table 4, the value of R^2^ was higher than 0.9, indicating that the Ostwald–de Wale model was suitable for evaluating the steady-state shear properties of the films in the gel state. All of the gels were pseudo-plastic fluids because the values of n were lower than 1.

The dynamic flow behaviors of gelled films were investigated at the linear viscoelastic region and the dissipation factor as a function of the angular frequency. Figure 7c,d shows that the dissipation factor (tanδ) of the films did not significantly change with the angular frequency from 0.1 to 100 rad/s, and its value was less than 1 (<0.15), indicating that the films in gel states had good long-term stability.

## 4. Conclusions

In this research, the κ-carrageenan film properties were improved by the addition of galactomannan (guar gum or locust bean gum). The results showed that galactomannan (GM) has good compatibility with κ-carrageenan and could enhance the thermal stability of κ-carrageenan-based films. As the GM content increased, the TS of the composite films had little variation and exceeded 19 MPa; moreover, the barrier performance of the composite films also improved. The GM with the higher M/G ratio could have led to the lower OP, SR and WVP of the GM–carrageenan composite films. Therefore, green and safe GM–carrageenan films with certain strength and barrier properties have potential in the food packaging industry. This study will promote the edible film to be applied in the future.

## Figures and Tables

**Figure 1 polymers-15-01751-f001:**
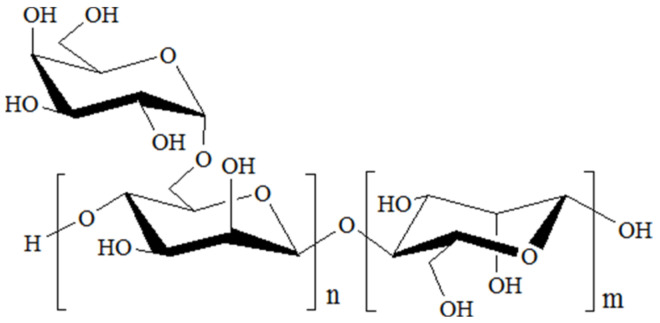
The typical structure of galactomannan.

**Figure 2 polymers-15-01751-f002:**
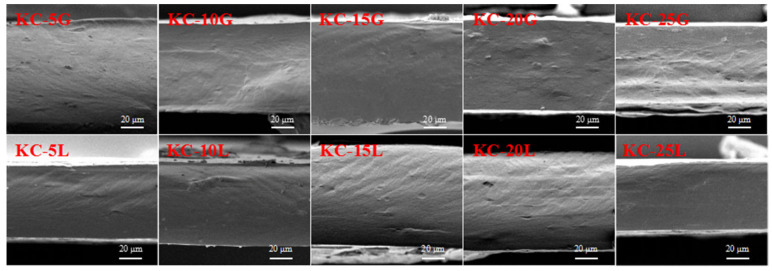
SEM images of carrageenan film containing GG and LG.

**Figure 3 polymers-15-01751-f003:**
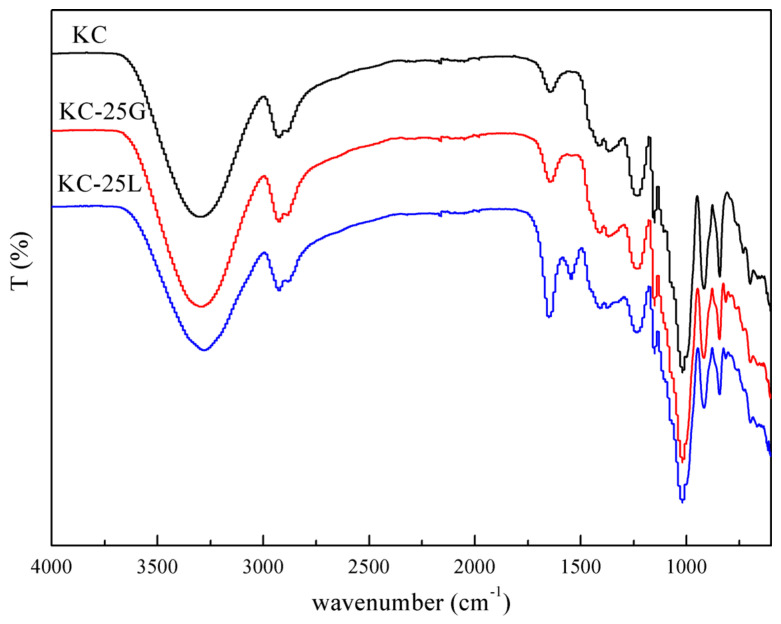
FTIR spectra of KC, KC-25G and KC-25L.

**Figure 4 polymers-15-01751-f004:**
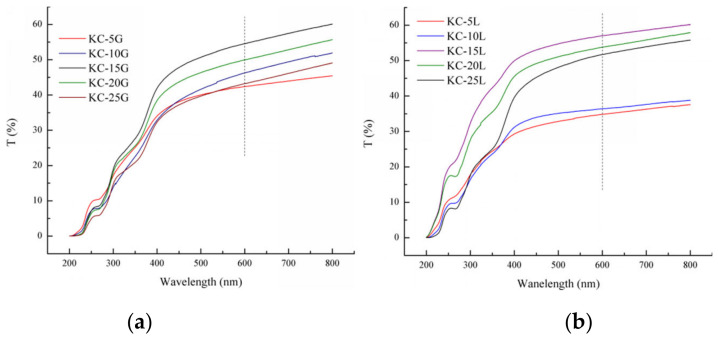
Light transmittance of the composite film containing GG (**a**) and LG (**b**).

**Figure 5 polymers-15-01751-f005:**
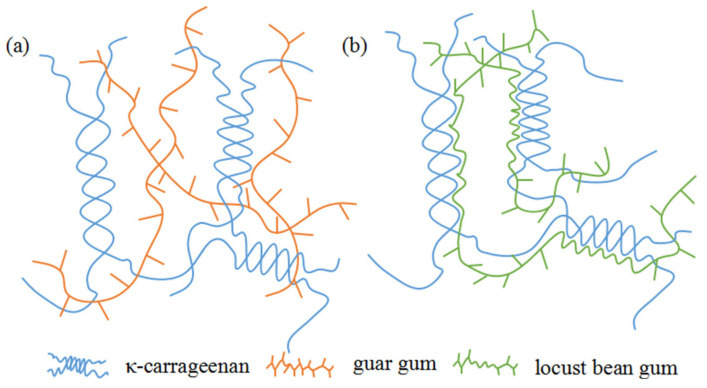
Binding mechanism diagram of KC g film (**a**) and KC-L film (**b**).

**Figure 6 polymers-15-01751-f006:**
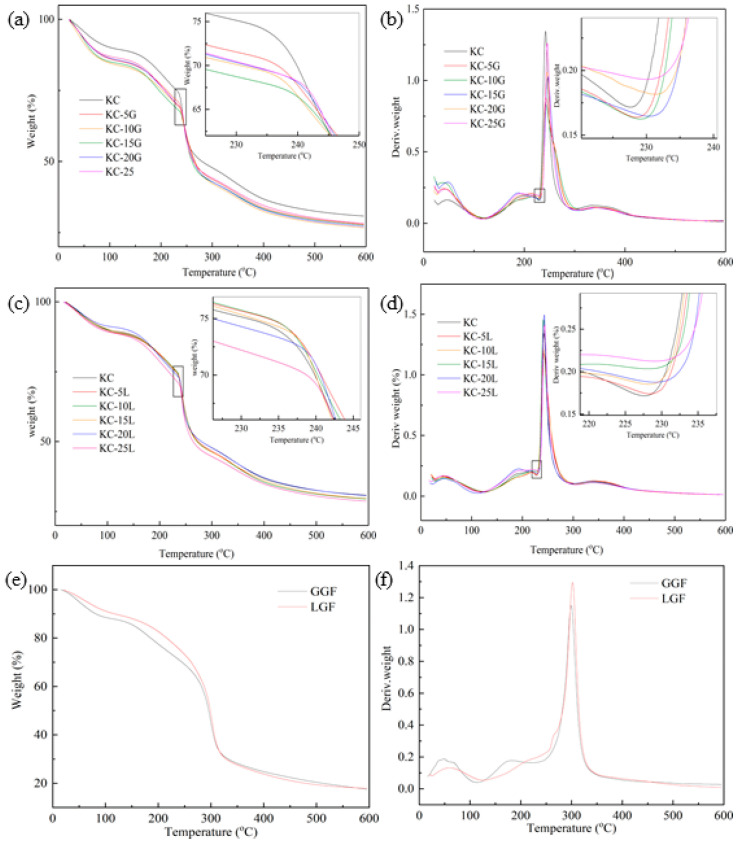
TGA (**a**,**c**,**e**) and DTG (**b**,**d**,**f**) curves of KC, KC-G, KC-L, GGF and LGF.

**Figure 7 polymers-15-01751-f007:**
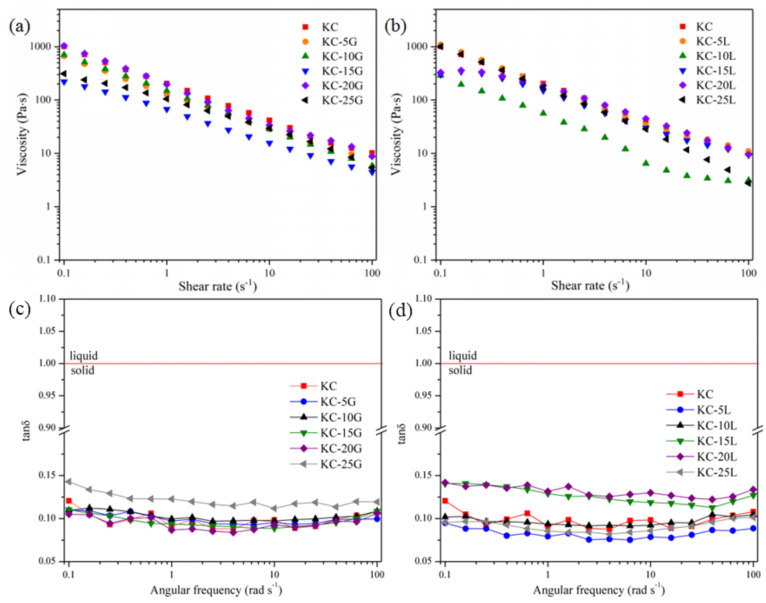
The steady-state shear properties (**a**,**b**) and the dissipation factor (**c**,**d**) of the films.

**Table 1 polymers-15-01751-t001:** TS and EB of KC, KC-G and KC-L films.

Sample	Thickness (mm)	TS (MPa)	EB (%)	OP (cm^3^ mm m^−2^ day^−1^ atm^−1^)
KC	0.107 ± 0.003 cd	21.775 ± 0.556 ab	37.150 ± 1.161 ce	3.247 ± 0.490 bd
KC-5G	0.100 ± 0.007 bc	21.380 ± 1.536 ab	32.300 ± 2.598 d	2.783 ± 0.074 bcd
KC-10G	0.105 ± 0.008 cd	21.298 ± 0.870 ab	23.900 ± 1.723 c	2.647 ± 0.147 bc
KC-15G	0.086 ± 0.000 a	21.880 ± 2.088 b	21.050 ± 1.143 b	2.360 ± 0.139 ab
KC-20G	0.096 ± 0.002 b	19.973 ± 1.013 ab	19.200 ± 0.849 ab	1.935 ± 0.074 a
KC-25G	0.096 ± 0.001 b	19.130 ± 1.179 a	18.150 ± 1.090 a	2.919 ± 0.017 cd
KC-5L	0.100 ± 0.002 b	22.933 ± 0.564 bc	36.700 ± 2.858 c	1.847 ± 0.070 a
KC-10L	0.097 ± 0.004 ab	23.988 ± 1.270 c	35.950 ± 1.873 c	1.799 ± 0.049 a
KC-15L	0.101 ± 0.003 bc	21.545 ± 0.809 ab	26.500 ± 1.212 b	1.595 ± 0.061 a
KC-20L	0.113 ± 0.005 d	21.555 ± 1.469 ab	21.000 ± 2.490 a	1.642 ± 0.196 a
KC-25L	0.091 ± 0.003 a	19.993 ± 0.851 a	20.600 ± 0.707 a	1.853 ± 0.145 a

Different lowercase letters in the same column indicate significant difference (*p* < 0.05).

**Table 2 polymers-15-01751-t002:** SR and WVP of KC, KC-G and KC-L.

Sample	SR (%)		WVP (g mm s^−1^ m^−2^ Pa^−1^ × 10^−11^)
	RH53%	RH75%	
KC	14.67 ± 2.08 b	25.09 ± 1.84 c	4.44 ± 0.09 ef
KC-5G	11.55 ± 1.47 ab	19.13 ± 0.07 b	3.81 ± 0.14 d
KC-10G	9.70 ± 0.44 ab	18.83 ± 0.92 b	3.68 ± 0.11 c
KC-15G	9.20 ± 1.02 ab	17.35 ± 1.17 ab	3.60 ± 0.10 c
KC-20G	7.44 ± 1.27 a	16.88 ± 1.72 ab	3.19 ± 0.08 b
KC-25G	6.10 ± 2.44 a	13.93 ± 1.34 a	3.08 ± 0.07 a
KC-5L	9.72 ± 0.46 a	18.73 ± 1.19 b	3.56 ± 0.04 e
KC-10L	8.21 ± 1.17 a	17.00 ± 0.80 b	3.36 ± 0.07 d
KC-15L	7.40 ± 1.51 a	16.70 ± 1.55 b	3.08 ± 0.07 c
KC-20L	6.61 ± 1.16 a	15.82 ± 1.15 ab	2.98 ± 0.08 b
KC-25L	5.94 ± 1.23 a	11.24 ± 2.09 a	2.35 ± 0.06 a

Different lowercase letters in the same column indicate significant difference (*p* < 0.05).

**Table 3 polymers-15-01751-t003:** Decomposition parameters of KC, KC-G, KC-L, GGF and LGF.

Sample	T_O_	T_m_	T_E_
KC	227.43	242.31	298.38
KC-5G	228.19	243.13	307.46
KC-10G	228.91	242.84	312.18
KC-15G	230.12	246.98	303.56
KC-20G	231.21	245.59	310.06
KC-25G	229.84	245.73	307.85
KC-5L	228.49	242.60	304.25
KC-10L	228.33	241.80	302.76
KC-15L	228.43	241.76	305.10
KC-20L	229.28	243.15	299.29
KC-25L	229.07	243.83	305.32
GGF	234.97	299.44	361.48
LGF	231.47	301.92	361.19

T_O_ is the initial decomposition temperature of the polysaccharide. T_m_ is the temperature of maximum decomposition rate. T_E_ is the end temperature of the main decomposition of polysaccharide.

**Table 4 polymers-15-01751-t004:** The Ostwald–de Wale model parameters of the films.

Sample	Ostwald-de Wale		
	K	n	R^2^
KC	197.353 ± 2.728	0.2956 ± 0.007	0.99939
KC-5G	130.905 ± 1.146	0.297 ± 0.005	0.99975
KC-10G	149.079 ± 0.860	0.326 ± 0.003	0.99988
KC-15G	65.673 ± 1.319	0.462 ± 0.011	0.99717
KC-20G	196.737 ± 2.272	0.277 ± 0.006	0.99961
KC-25G	102.195 ± 2.143	0.515 ± 0.011	0.99578
KC-5L	197.413 ± 2.318	0.254 ± 0.006	0.99963
KC-10L	53.858 ± 1.240	0.277 ± 0.012	0.99846
KC-15L	142.467 ± 11.551	0.578 ± 0.046	0.91772
KC-20L	166.968 ± 10.988	0.614 ± 0.038	0.92924
KC-25L	178.615 ± 1.934	0.246 ± 0.006	0.9997

## Data Availability

Not applicable.

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
