# Peer review of "A Contribution to Improve Barrier Properties and Reduce Swelling Ratio of κ-Carrageenan Film from the Incorporation of Guar Gum or Locust Bean Gum"

_polymers, 2023, doi:10.3390/polym15071751_

Round 1
Reviewer 1 Report
This paper deals with the addition of galactomannan to improve the properties of carrageenan-based films.
The following comments should be addressed.
INTRODUCTION
Line 48: what is M/G ratio? What is M and G? You did not identify them before.
Line 50-52: you cited only two works and you did not report their findings. You should improve the background of your research and touch upon the main results.
Line 56: as regards the use of glycerin, you should describe better its impact and cite recent literature about bio-based films added with glycerin (such as “Stability of film-forming dispersions: Affects the morphology and optical properties of polymeric films”, 2021; “Multiple light scattering as a preliminary tool for starch-based film formulation”, 2022; “Starch/pectin‐biobased films: How initial dispersions could affect their performances”, 2022).
RESULTS
You should compare your results with other film characteristics, previously studied (possibly carrageenan-based film, but also other biobased films such as starch-films). The discussion is not sufficient to contextualize your work in the inherent literature.
CONCLUSIONS
Why do carrageenan films have potential application in food packaging? Which characteristics are reguired for food packaging?
Moreover, extensive editing of English language and style is required.
e.g
Title: “the contribution to improving … and reducing…” should be replaced by “the contribution to improve …and reduce…”. Then, “corporation” is not an appropriate term (because it refers to other topics”) and should be replaced by “incorporation”
Line 23: “The Recently, plastic brings great…” has no sense!
Line 25: “only 3% of it” should be replaced by “only 3% of them”
Line 26: “may has” should be replaced by “should have”
Line 28: “it may contain” should be replaced by “they may contain” because it is referred to “plastics”.
Line 28: “it is urgently” should be replaced by “it is urgent”
and so on through the entire manuscript. Therefore, this paper could be considered for publication after an extensive revision.
Reviewer 2 Report
The thermal stability study can be improved.
First, according to DTG curves, there are at least four overlapping mass loss processes.
The assignment of moisture, glycerol and polysaccharide to the three stages of mass loss described in the article should be justified with references to other works or experimental results.
Although the highest calculated onset corresponds to GG, it does not mean that GG has the highest stability. In fact, Figurte 6c shows that LG has a higher stability than GG.
I think the slight differences observed in thermal stability do not justify to say, in conclusions, that “the addition of GM made the κ-carrageenan based films have potential application in food packaging”.
Regarding the sentence “Rheological behavior of the films in the gel state”, it should be explained whether the gel film sample is the same as the one obtained as described in the sample preparation section (dried for ~24 h at 50 °C).
This sentence “The dynamic flow behaviors of gelled films were investigated at the linear viscoelastic region” should be based on a test. How was the linear range determined?
This sentence in the Conclusions section does not seem convincing: The composite films containing GG with higher branching had lower EB because the intermolecular fluidity of polysaccharides is more difficult.
Fig 6 (a) right. The y-axis label is missing
The last Figure does not have a caption.
Figure number seems to be wrong in this text “Figure 7. shows that the dissipation factor…”
Reviewer 3 Report
The manuscript has investigated the influence of guar gum or locust bean gum on the physicochemical properties of κ-carrageenan films. However, there are several published papers with similar topics and the novelty of this work is not highlighted. Furthermore, the presented values in the tables are not statistically analyzed and the results are poorly discussed. The conclusion is not appropriate and is similar to the abstract.
Round 2
Reviewer 1 Report
Please, check the reference numbers in the Introduction: in line 30 you cited the Reference [2], but then the reference started from the number [9] from the Line 33; also in Lines 46-51, you cited the papers [21], [23] and [25] but not [24]. Therefore, where did you cite in the text [3-8], [22] and [24], that you reported in the Reference section at the end of the paper?
Section 3.4: you cited Figures 5(a) and 5 (b), but they are not present in the paper.
You should improve the description of figures and tables:
- Tables 1-2: what are the letters a, b, c, d, e, f? What do they indicate?
- Figure 6: specify the letters a-f in the caption
- Improve the conclusions with the comment that you reported as answer to reviewer's comments.
Please, a further revision of English editing is necessary (just as an examples Section 2.4.8 "Data were" and not "data was"; and so on)
Reviewer 2 Report
My concerns have been adequately addressed in the current version and, thus, I believe it is suitable to be accepted.
Reviewer 3 Report
The manuscript is acceptable.
